# Increased Prevalence of Myocardial Infarction and Stable Stroke Proportions in Patients with Inflammatory Bowel Diseases in Quebec in 1996–2015

**DOI:** 10.3390/jcm11030686

**Published:** 2022-01-28

**Authors:** Petra Anna Golovics, Christine Verdon, Panu Wetwittayakhlang, Christopher Filliter, Lorant Gonczi, Gustavo Drügg Hahn, Gary E. Wild, Waqqas Afif, Alain Bitton, Talat Bessissow, Paul Brassard, Peter L. Lakatos

**Affiliations:** 1Division of Gastroenterology, McGill University Health Centre, Montreal, QC H3G 1A4, Canada; golovics.petra@gmail.com (P.A.G.); cver3698@uni.sydney.edu.au (C.V.); wet.panu@gmail.com (P.W.); gustavodhahn@gmail.com (G.D.H.); drgarywild@gmail.com (G.E.W.); waqqas.afif@mcgill.ca (W.A.); abit66@gmail.com (A.B.); talat.bessissow@gmail.com (T.B.); 2Department of Gastroenterology, Hungarian Defence Forces, Medical Centre, H-1062 Budapest, Hungary; 3Unit of Gastroenterology and Hepatology, Division of Internal Medicine, Faculty of Medicine, Prince of Songkla University, Songkhla 90110, Thailand; 4Centre for Clinical Epidemiology, Lady Davis Research Institute, Jewish General Hospital, Montreal, QC H3T 1E2, Canada; christopher.filliter@ladydavis.ca (C.F.); paul.brassard@ladydavis.ca (P.B.); 51st Department of Medicine, Semmelweis University, H-1083 Budapest, Hungary; lorantgonczi@gmail.com; 6Graduate Course Sciences in Gastroenterology and Hepatology, School of Medicine, Universidade Federal do Rio Grande do Sul, Porto Alegre 90035-002, Brazil; 7Department of Medicine, McGill University, Montreal, QC H3A 1A1, Canada

**Keywords:** inflammatory bowel disease, myocardial infarction, stroke, prevalence, incidence, risk factor

## Abstract

Background: Chronic inflammatory diseases are linked to an increased risk of atherothrombotic events, but the risk associated with inflammatory bowel disease (IBD) is controversial. We therefore examined the risk of and risk factors for myocardial infarction (MI) and stroke in IBD patients. Methods: We used the public health administrative database from the Province of Quebec, Canada, to identify IBD patients newly diagnosed between 1996 and 2015. The incidence and prevalence of MI and stroke in IBD patients were compared to those for the Canadian population. Results: A cohort of 35,985 IBD patients was identified. The prevalence but not incidence rates of MI were higher in IBD patients (prevalence: 3.98%; incidence: 0.234) compared to the Canadian rates (prevalence: 2.0%; incidence: 0.220), while the prevalence and incidence rates of stroke were not significantly higher in the IBD patients (prevalence: 2.98%; incidence: 0.122, vs. Canadian rates: prevalence: 2.60%; incidence: 0.297). We identified age, female gender, hyperlipidemia, diabetes, and hypertension (*p* < 0.001 for each) as significant risk factors associated with MI and stroke in IBD. Exposure to biologics was associated with a higher incidence of MI (IRR: 1.51; 95% CI: 0.82–2.76; *p* = 0.07) in the insured IBD population. Conclusions: An increased prevalence but not incidence of MI and no increased risk of stroke were identified in this population-based IBD cohort.

## 1. Introduction

Inflammatory bowel diseases (IBDs) are chronic, progressive, and disabling conditions mainly affecting young adults. IBD substantially impacts patients’ physical health, social functioning, and quality of life, thus contributing to the high health–economic burden associated with the disease. While cardiovascular diseases (CVDs) remain the primary cause of mortality and an essential public health concern in the developed world, a diverse set of inflammatory conditions (e.g., rheumatoid arthritis, systemic lupus erythematosus, and ankylosing spondylitis) has previously been linked with the progression of atherogenesis and, consequently, increased cardiovascular outcomes [1]. However, a national cohort from Australia suggested that anti-TNF therapy was associated with a reduction in major cardiovascular events in rheumatoid arthritis, psoriatic arthritis, and ankylosing spondylitis [2]. Moreover, two recent meta-analysis showed a decreased incidence and mortality rate in psoriasis and psoriatic arthritis patients treated with anti-TNF therapy [3,4]. By contrast, the data pertaining to IBD are more limited and less robust.

With respect to Canada, Bernstein et al. confirmed, in a national study that included an assessment of all hospitalizations across Canada, that both Crohn’s disease (CD) and ulcerative colitis (UC) are associated with a three- to four-fold increased risk for venous thrombosis including pulmonary embolism [5]. However, data regarding the impact of IBD on arterial embolism were less convincing. Interestingly, earlier data derived from the same provincial (i.e., Manitoba) [6] epidemiological database showed an increased risk of arterial thromboembolic diseases for all IBD patients, regardless of diagnosis or gender (incidence rate ratio (IRR), 1.26; 95% confidence interval (CI), 1.11–1.44), and cerebrovascular disease risk elevation was demonstrated only in CD patients (IRR, 1.32; 95% CI, 1.05–1.66). 

A meta-analysis reported by Fumery et al. [7] showed no increase in cardiovascular death, despite an increase in the risk of ischemic heart disease (IHD) (relative risk (RR), 1.35 (1.19–1.52)) in the IBD cohort. However, many studies demonstrate a positive correlation.

Considering the limited and partly conflicting data, our aim was to evaluate the cardiovascular and cerebrovascular outcomes and their risk factors in a defined population-based inception IBD cohort using a health administrative database of the province of Quebec, Canada. 

## 2. Materials and Methods

### 2.1. Study Design and Data Collection

We used the population-based inception IBD cohort using the health administrative database of the province of Quebec. A more detailed description is available in our previous publication [8]. Administrative data on demographics (age and gender), IBD diagnosis (UC vs. CD), cardiovascular risk factors (hypertension, dyslipidemia, diabetes, and pertinent medications in defining the condition), and outcomes (ischemic heart disease, myocardial infarct, cerebrovascular stroke, and cardiovascular death) were obtained from the public health administrative database from the Province of Quebec (Régie d’Assurance Santé du Quebec or RAMQ) and MedECHO database for the incident IBD patients diagnosed between 1996 and 2015. 

The incidence and prevalence of stroke and MI were defined using ICD codes found in primary and secondary care visits or admissions [8].

We used the following ICD codes for MI—(ICD-10) I21–I22 and (ICD-9) 410—and for ischemic heart disease (including MI), we used (ICD-10) I20–25 and (ICD-9) 410–414. In the case of stroke, we used (ICD-10) I63–64 and (ICD-9) 433–438; if the stroke included subarachnoid and intracerebral hemorrhage, then (ICD-10) I60–I64 and (ICD-9) 431–438 were used.

This IBD cohort was compared to the available Canadian control population data (the rates were available for 2012–2013) [9,10,11,12] adjusted for age and gender, to evaluate the incidence and prevalence of cardiovascular outcomes. In addition, we performed a risk analysis and compared major risk factors accessible within the administrative database (hypertension, dyslipidemia, and diabetes) to those for the general population. Furthermore, we assessed the association between cardiovascular outcomes and IBD activity. 

The STROBE statement checklist is available online as Appendix A.

### 2.2. Statistical Analysis

#### 2.2.1. Cardiovascular Outcome Analysis

We characterized an IBD cohort (UC and CD) and a non-IBD cohort adjusted for age and gender from the RAMQ administrative database. We obtained the incidence rate (IR) and prevalence odds ratio (OR) for the IBD population for each of the following: ischemic heart disease (IHD), myocardial infarction (MI), cardiovascular disease, and stroke. We compared the IR and OR obtained to those for a non-IBD cohort adjusted for age and gender, and available census data (Quebec +/− Canada rates, as available). In addition, we performed a sub-analysis in all IBD patients and UC versus CD patients. We performed an age-related subgroup analysis according to the two categories of <40- and ≥40-year-old patients and another according to gender (male and female). Finally, we compared the incidence and prevalence of both outcomes in the IBD cohort with the Canadian population data.

#### 2.2.2. Risk Factor Analysis 

The risk factors (diabetes, hypertension, and hyperlipidemia) were collected and calculated in the total cohort as well as in the insured cohort, including information regarding medical therapy. The data were analyzed in both the total population (called the newly diagnosed IBD cases subgroup) and in a sensitivity analysis; we also assessed the importance of risk factors in the insured population, meaning in patients with public insurance coverage (called the IBD cases with RAMQ prescription insurance subgroup). Data on private drug dispensation were not available.

In that analysis, the prevalence was defined as having had any outcome prior to 1 January 2014. We used that date to obtain data that were comparable to the Canadian rates. Incidence was defined using a subgroup of the IBD cohort that survived with no outcome at least 2 years after cohort entry. This ensured that the cases were incident. The outcome could occur any time after that. The person-time for incidence analysis was computed from cohort entry plus two years, until the end of the study. We put no restriction on a calendar year for the incidence analysis. 

For the prevalence analysis, the rates were computed by dividing the total number of patients with an outcome, as defined above, by the total number of persons at risk. The rates were compared with Canadian rates using a Chi-square test for proportions. For the incidence analysis, we fitted a Poisson regression model for the number of outcomes, using the person-time as an offset. The only variable in the model consisted of an indicator of the IBD cohort or Canadian rates. The rates were computed for 100 person-years; we obtained *p*-values to compare between the IBD cohort outcome rates and Canadian rates. 

The prevalence and the incidence analyses were further stratified by IBD type (CD/UC), by gender, and by age (<40/≥40 years old).

#### 2.2.3. Comorbidity Analysis

In the comorbidity analysis, the prevalence for an outcome was defined as having had an outcome at any time, either before cohort entry or during follow-up, with no restriction on calendar year, while we defined incidence using a two-year buffer period where patients had to have at least 2 full years without the outcome. For the incidence analysis, the person-time was computed from the maximum date between cohort entry and 1 January 1998 (which corresponds to 1 January 1996, plus two years) to the end of the study, as we excluded any patient with outcomes that occurred before 1 January 1998—these patients not having a 2-year outcome-free buffer.

The prevalent and incident outcomes were modelled using a logistic regression model and a Poisson model, respectively. For the prevalent outcomes, we first fitted logistic regressions adjusted for age and gender along with one comorbidity of interest (e.g., diabetes in the first iteration, and then hyperlipidemia and hypertension). In the univariate analysis, any of the 3 comorbidities that presented a significant coefficient (*p*-value < 0.01) were kept and added to a multivariate model along with age and gender. For all the models and for each variable included, the odds ratio, 95% CIs, and *p*-values were computed. For the Poisson model, the outcome rate for 1000 person-years was fitted, first adjusting for age, gender, and every comorbidity one at a time (diabetes, hyperlipidemia, and then hypertension). Each of the significant comorbidity variables (*p*-value < 0.1) was subsequently added to a multivariate Poisson regression model along with age and gender. For all the models and for each variable included, the incidence rate ratios, 95% Cis, and *p*-values were computed.

The analyses were further stratified by IBD type (CD and UC) for both the prevalence and incidence analyses. 

#### 2.2.4. Disease Severity Analysis

The maximal therapeutic step was used as a proxy for disease activity and severity. The therapeutic groups were separated into three categories: 5-aminosalicylic acid (5-ASA), immunosuppressants (ISs)/corticosteroids, and biologics. The incident rate ratio (IRR) for each cardiovascular outcome (IHD, MI, cardiovascular disease, and stroke) was obtained for each group/category in time-varying analyses. 

#### 2.2.5. Ethical Considerations 

The study was approved by the McGill University Health Centre institutional review board and ethics committee (REB no. 2013-110, 2019-5272). The research protocol conforms to the ethical guidelines of the 1975 Declaration of Helsinki and local regulations. 

## 3. Results 

A total of 35,985 newly diagnosed IBD patients were identified between 1996 and 2015 in the RAMQ database, with 10,915 in the insured cohort (Table 1).

The majority of the patients had Crohn’s disease (57.3%), were female (53.9%), and were diagnosed over the age of 40 years (58.2%).

In the insured subgroup, a higher proportion of the patients were diagnosed over 40 years (64.7%), were female (55.3%), and had Crohn’s disease (57.6%). The maximum therapeutic step before either myocardial infarction or stroke was nearly the same for 5-ASA, 15.7%/15.7%, and for biologics, 12.8%/12.9%. The majority of the patients used immunosuppressive drugs or steroids. Considering the major comorbidities, we found that 9.5% of the IBD patient had diabetes, 24.9% had hypertension, and 24.5% had hyperlipidemia.

### 3.1. Cardiovascular Outcomes

Analyzing the whole patient population, the 35,985 newly diagnosed IBD patients, and the non-IBD cohort adjusted for age and gender from the RAMQ administrative database showed that the prevalence but not incidence rates of MI were higher in IBD (prevalence at the end of 2013: 3.98%; incidence: 0.234/100,000/yr) compared to the background Canadian rates (prevalence in 2012–2013: 2.0%; incidence: 0.220/100,000/yr). Meanwhile, the prevalence and incidence rates of stroke were not significantly higher in IBD (prevalence in 2012–2013: 2.98%; incidence: 0.122/100,000/yr) vs. the Canadian population (prevalence in 2012–2013: 2.60%; incidence: 0.297/100,000/yr). 

### 3.2. Risk Factors

The risk factors were collected and analyzed in the total cohort as well as in the insured cohort, including specific information regarding medical therapy. We identified older age, female gender, hyperlipidemia, and hypertension (*p* < 0.001 for each) as risk factors for developing MI and stroke in both CD and UC in the logistic-regression-based prevalence models (Table 2, Table 3, Table 4 and Table 5), but not in all the Poisson-regression-based incidence models. Diabetes was identified as an additional risk factor for MI in CD and stroke in UC in both models (available online only, Appendix A). Using a sensitivity analysis, we assessed risk factors in the insured population. The results were in keeping with those obtained in the total population.

### 3.3. Disease Severity

The maximal therapeutic step was used as a proxy for disease activity and severity. In this analysis, we were able to use only the insured population data for 10,915 cases. Exposure to biologicals was associated with a higher incidence of MI compared to the non-treatment group (IRR: 1.51; 95% CI: 0.82–2.76; *p* = 0.07 for all drug exposure) in the insured IBD population in the time-varying model (Table 6).

## 4. Discussion

The major finding of the present study is an increased prevalence but not incidence of MI in this population-based IBD cohort from the province of Quebec. No increased risk of stroke was identified in this cohort of patients. The risk factors for both MI and stroke in IBD patients included age, female gender, hyperlipidemia, and hypertension. Exposure to biological therapy was associated with a tendency for a higher incidence rate ratio for MI in IBD (*p* = 0.07). Thus, the therapeutic step can be regarded as a disease-severity marker in administrative databases.

It is noteworthy that few studies have assessed this outcome in IBD. Osterman et al. [13] reported, in a United Kingdom population-based study, no increased risk of myocardial infarct in IBD patients compared to the general population. A further study by Kristensen et al. [14] investigated cardiovascular outcomes in a Danish population-based national registry in IBD patients and found that the overall risk of myocardial infarct (RR: 1.17 (95% CI: 1.05–1.31) and stroke (RR: 1.15 (1.04–1.27)) and cardiovascular death (RR: 1.35 (1.25–1.45)) was increased overall, and even more so in patients with active disease. In addition, a 2014 systematic review and meta-analysis by Singh et al. [15] from Mayo Clinic corroborated these findings with increased rates of both IHD (OR: 1.19; 95% CI: 1.08–1.31) and stroke (OR: 1.18; 95% CI: 1.09–1.27) in IBD patients, especially amongst women (26% vs. 5% for IHD; 28% vs. 11% for stroke). A striking finding was the increased risk within the younger cohort (an HR of 1.40 for IHD and HR of 1.84 for stroke) compared to patients over 50 years old. Interestingly, the traditional risk factors were not necessarily increased in the IBD versus non-IBD cohort. A more recent meta-analysis by Feng et al. [16] in 2017 confirmed an increased risk of IHD, with a RR of 1.244 (95% CI: 1.14–1.35). In the present study, we found an increased prevalence but not incidence of MI in IBD. It is a common perception that there has been a change in the sensitivity in the identification of myocardial necrosis since the introduction into clinical practice of high-sensitivity troponins, and this may reduce the incidence of diagnoses of (unstable) angina while increasing the diagnoses of NSTEMI. Of note, this phenomenon was not captured in the present administrative registry.

More recent studies include a study by Baena-Diez et al. [17], which evaluated the increased risk of CVD in a Catalonian electronic database of general practitioners and showed a positive association with IBD (HR: 1.18 (1.06–1.32)). Additionally, Kirchgesner et al. [18] from France confirmed that both CD and UC patients have a statistically significant overall increased risk of acute arterial events (IHD, stroke, and peripheral vascular disease (PVD)) with standardized incidence ratios (SIRs) of 1.35 (95% CI: 1.30–1.41) and 1.10 (95% CI: 1.06–1.13), respectively. Females were at higher risk of adverse outcomes. This is similar to the results from the present study. The patients with the highest risk were females with CD, suggesting that the conventional protective effect of female gender is probably counteracted by the chronic inflammation in IBD, and this effect may be more pronounced in females compared to males.

Among patients without traditional cardiovascular risk factors, an increased risk of acute arterial events was reported in CD (SIR: 1.26 (1.19–1.34)) versus in UC (SIR: 0.96 (0.92–1.01)). However, the risk increased in parallel with an increasing number of risk factors. However, the data on the CVD risk in IBD are partly conflictive. An early meta-analysis by Dorn et al. analyzed 4532 CD and 9533 UC patients’ data and showed no increased CVD mortality rates (a standardized mortality ratio for CD of 1.0 (95% CI: 0.8–1.1) and for UC of 0.9 (95% CI: 0.8–1.0)) [19]. 

One of the aims of the present study was to study the relative importance of the conventional CVD risk factors in IBD patients. Thus, we aimed to identify if the conventional risk factors assessed in this study (no data on smoking were available) were more prevalent in IBD patients with MI or stroke outcomes, but we did not directly compare the prevalence of CVD risk factors with that in non-IBD controls or the general population.

In the present study, older age, female gender, hyperlipidemia, hypertension, and diabetes (*p* < 0.001 for each) were the significant risk factors for developing MI in IBD and both CD and UC. Of note, the prevalence of traditional CV risk factors was not more common, although it was numerically lower in patients with IBD compared to that reported in the general population (hypertension: 16.4% vs. 22.6%; diabetes: 6.3% vs. 8.8%) in other studies [20,21]. Of note, a recent study by Emanuel et al. [22] compared differences in CVD risk factor assessment and management among patients with RA and IBD in a primary care setting. They found that the assessment and the treatment of vascular risk in RA and IBD patients in primary care are suboptimal, particularly with reference to CVD risk calculation. However, patients with IBD were 26% more likely (IRR: 1.26; 95% CI: 1.16–1.38) to have CVD risk factors measured compared with matched controls. The difference declined to 10% (IRR: 1.10; 95% CI: 1.05–1.15) over 5 years of follow-up. The authors concluded that more precise studies were required to assess the CVD risk in IBD. 

In terms of stroke risk, three meta-analyses have been published. Yuan et al. [23] performed a meta-analysis of eight studies and reported an RR of 1.31 (1.16–1.47) between IBD and the risk of stroke, after adjusting for established cardiovascular risk factors. This risk is substantial for women (RR: 1.46 (1.12–1.91)) compared to men (RR: 1.23 (1.04–1.45)). The Xiao et al. [24] meta-analysis included eight studies, seven of which are the same as those reported by Yuan et al.; hence, as expected, both meta-analyses have reported similar results and conclusions. Xiao et al., however, performed an ethnicity-specific analysis, demonstrating a higher risk of stroke in Asian patients with IBD (HR: 1.6 (1.04–2.48) compared to Caucasian patients with IBD (HR: 1.23 (1.12–1.34). Fumery et al. [7] aimed to evaluate the risk of thromboembolism in IBD, including stroke. They reported a negative association, with a RR of 0.79 (0.51–1.23), but used only three studies in their analysis. In the present study, the prevalence and incidence rates of stroke were not significantly higher in IBD. 

Nevertheless, the risk factors for developing stroke in the overall IBD group, the insured IBD group, and the CD or UC population were age, diabetes, and hypertension (*p* < 0.001 for each). In the IBD group, female gender was associated with stroke prevalence but not with incidence. Hyperlipidemia was associated in IBD, CD, and UC with stroke prevalence but not incidence.

Disease activity has been linked to cardiovascular outcomes, and this may represent an independent risk factor as shown in the French IBD population [25]. These data underscore the need to ensure optimal medical therapy to minimize inflammation severity and duration. In concordance, the data presented here in the insured patient population show that exposure to biologicals is associated with a higher incidence of MI compared to the non-exposed group (IRR: 1.51; 95% CI: 0.82–2.76; *p* = 0.07 for all drug exposure) in the time-dependent model.

The strength of this study stems from the large, well-characterized, population-based IBD cohort and methodology from the provincial administrative database for the Province of Quebec (1996–2015). We used well-defined administrative criteria to identify IBD patients as well as outcomes and risk factors. We evaluated both the incidence and prevalence of outcomes in IBD and performed a subgroup analysis for CD and UC as well as assessing the comorbidities and risk factors. There are some limitations to our cohort. No data on smoking status could be obtained. In addition, the Quebec drug plan for biologics therapy is two-fold, with public RAMQ or private insurance coverage; the latter was not captured in our data set. Of note, 41.8% of prescribed drug spending is paid for by provincial healthcare plans across Canada [26]. To decrease the bias from these limitations, we performed multiple sensitivity analysis to empower the strength of our analysis. 

In conclusion, in this original, long-term, population-based, observational study from Quebec, Canada, we used a sophisticated statistical approach and have demonstrated that the prevalence but not incidence rates of MI were higher in IBD compared to the Canadian background population. By contrast, we did not uncover a higher prevalence or incidence of stroke in IBD. In addition, we identified older age, female gender, hyperlipidemia, diabetes, and hypertension as risk factors for developing MI in IBD, CD, and UC. Finally, in the insured IBD population, the exposure to biological therapy was associated with a tendency toward a higher incidence rate ratio for MI in IBD, further confirming that therapeutic steps can be regarded as a disease-severity marker in administrative databases. 

## Figures and Tables

**Table 1 jcm-11-00686-t001:** Characteristics of IBD patients and general population between 1996 and 2015 in Quebec, Canada.

	Newly Diagnosed IBD Cases (*n* = 35,985)	IBD Cases with RAMQ Prescription Insurance (*n* = 10,915)
*Age category*	Age < 40 yrs—41.7% Age > 40 yrs—58.2%	Age < 40 yrs—35.3% Age > 40 yrs—64.7%
*Gender*	Female—53.9% Male—46.1%	Female—55.3% Male—44.7%
*IBD*	CD—57.3% UC—38.9% Other—3.7%	CD—57.6% UC—38.7% Other—3.7%
*Hyperlipemia*	13.8%	9.5%
*Hypertension*	16.4%	24.9%
*Diabetes*	6.3%	24.5%
**Maximum therapeutic step before myocardial infarction**		
5-ASA	9.7%	15.7%
Biologics	9.2%	12.8%
Steroid/immunosuppressant	31.8%	53.9%
**Maximum therapeutic step before stroke**		
5-ASA	9.7%	15.7%
Biologics	9.2%	12.9%
Steroid/immunosuppressant	31.8%	53.9%

**Table 2 jcm-11-00686-t002:** Association between risk factors and myocardial infarction prevalence in total IBD population.

IBD/MI Prevalence Total Population *^n^*	Univariate Analysis			Multivariate Analysis		
Variables	OR	95% CI	*p*-Value	OR	95% CI	*p*-Value
*Age*	1.077	(1.073–1.081)	<0.0001	1.046	(1.042–1.051)	<0.0001
*Gender*	2.079	(1.861–2.323)	<0.0001	2.033	(1.813–2.279)	<0.0001
*Diabetes*	2.278	(1.995–2.601)	<0.0001	1.346	(1.164–1.557)	<0.0001
*Hypertension*	3.559	(3.122–4.056)	<0.0001	2.418	(2.091–2.797)	<0.0001
*Hyperlipidemia*	3.497	(3.103–3.941)	<0.0001	2.805	(2.455–3.204)	<0.0001

Note: Univariate analysis adjusted for age and gender. OR: odds ratio; CI: confidence interval; *^n^* = 35,985.

**Table 3 jcm-11-00686-t003:** Association between risk factors and stroke prevalence in total IBD population.

IBD/Stroke Prevalence Total Population *^n^*	Univariate Analysis			Multivariate Analysis		
Variables	OR	95% CI	*p*-Value	OR	95% CI	*p*-Value
*Age*	1.086	(1.082–1.091)	<0.0001	1.064	(1.059–1.069)	<0.0001
*Gender*	1.248	(1.096–1.42)	0.0008	1.23	(1.078–1.403)	0.0021
*Diabetes*	1.792	(1.532–2.096)	<0.0001	1.275	(1.073–1.514)	0.0057
*Hypertension*	2.63	(2.26–3.062)	<0.0001	2.275	(1.92–2.697)	<0.0001
*Hyperlipidemia*	1.879	(1.64–2.152)	<0.0001	1.661	(1.425–1.936)	<0.0001

Note: Univariate analysis adjusted for age and gender. OR: odds ratio; CI: confidence interval; *^n^* = 35,985.

**Table 4 jcm-11-00686-t004:** Association between risk factors and myocardial infarction incidence in total IBD population.

IBD/MI Incidence Total Population *^n^*	Univariate Analysis			Multivariate Analysis		
Variables	IRR	95% CI	*p*-Value	IRR	95% CI	*p*-Value
*Age*	1.071	(1.066–1.076)	<0.0001	1.058	(1.053–1.064)	<0.0001
*Gender*	1.831	(1.57–2.135)	<0.0001	1.800	(1.542–2.101)	<0.0001
*Diabetes*	1.904	(1.556–2.328)	<0.0001	1.514	(1.227–1.868)	0.0001
*Hypertension*	1.992	(1.662–2.288)	<0.0001	1.666	(1.372–2.022)	<0.0001
*Hyperlipidemia*	1.708	(1.437–2.029)	<0.0001	1.368	(1.138–1.644)	0.0008

Note: Univariate analysis adjusted for age and gender. IRR: incidence rate ratio; CI: confidence interval; *^n^* = 35,985.

**Table 5 jcm-11-00686-t005:** Association between risk factors and stroke incidence in total IBD population.

IBD/Stroke Incidence Total Population *^n^*	Univariate Analysis			Multivariate Analysis		
Variables	IRR	95% CI	*p*-Value	IRR	95% CI	*p*-Value
*Age*	1.076	(1.069–1.083)	<0.0001	1.068	(1.060–1.076)	<0.0001
*Gender*	1.195	(0.976–1.464)	0.084	1.193	(0.973–1.463)	0.0886
*Diabetes*	1.602	(1.213–2.116)	0.0009	1.4589	(1.096–1.941)	0.0096
*Hypertension*	1.547	(1.218–1.961)	0.0003	1.4552	(1.139–1.858)	0.0026
*Hyperlipidemia*	1.177	(0.934–1.483)	0.1666	NA	NA	NA

Note: Univariate analysis adjusted for age and gender. IRR: incidence rate ratio; CI: confidence interval; *^n^* = 35,985.

**Table 6 jcm-11-00686-t006:** Association between risk factors and myocardial infarction incidence in the insured population.

IBD/MI Incidence Insured Population *^n^*	Univariate Analysis			Multivariate Analysis		
Variables	IRR	95% CI	*p*-Value	IRR	95% CI	*p*-Value
*Age*	1.059	(1.048–1.069)	<0.001	1.047	(1.035–1.059)	<0.0001
*Gender*	1.521	(1.143–2.024)	0.004	1.503	(1.127–2.00)	0.006
*Diabetes*	1.774	(1.239–2.539)	0.002	1.422	(0.976–2.071)	0.067
*Hypertension*	1.988	(1.448–2.731)	<0.0001	1.759	(1.258–2.462)	0.001
*Hyperlipidemia*	1.479	(1.093–2.001)	0.011	1.177	(0.852–1.625)	0.322
*5-ASA*	0.7686	(0.531–1.111)	0.1612	0.76	(0.529–1.105)	0.153
*Biologics*	1.5612	(0.852–2.859)	0.04	1.51	(0.82–2.76)	0.07
*Steroid*/IS	1.2757	(0.899–1.81)	0.1726	1.24	(0.871–1.756)	0.233

Note: Univariate analysis adjusted for age and gender. IRR: incidence rate ratio; CI: confidence interval; IS: immunosuppressant; *^n^* = 10,915.

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
