# Peer review of "Increased Prevalence of Myocardial Infarction and Stable Stroke Proportions in Patients with Inflammatory Bowel Diseases in Quebec in 1996–2015"

_jcm, 2022, doi:10.3390/jcm11030686_

Round 1

Reviewer 1 Report

Dear Golovics and colleagues, you performed an analysis of the incidence and prevalence of myocardial infarction and stroke in patients with IBD, in comparison with the general population. They used  a health administrative database of the province of Quebec to obtain patient informations.

The topic is certainly of scientific interest: the prevalence of IBD in the general population is high, the data regarding the incidence of thromb-embolic diseases to date are conflicting and the results obtained could be a guide for a more correct risk management in this population.

However some considerations: 

  • it is not clear how the risk factors (diabetes mellitus, dyslipidemia and hypertension) were defined: with an established diagnosis? by analysis of current therapy? As commented by the authors, the data on smoking is also missing.
  • the distinction between the two cohorts (newly diagnosed IBD cases vs IBD cases with RAMQ prescription insurance  is not clear. Furthermore, the differences between patient characteristics are not commented (a higher incidence of CV risk factors in the second population is evident)
  • IBD patients were compared  with a age-matched population. However, the age of onset of the CV disease could be different in the two populations as well as the age of onset of the various risk factors, as is well known in the context of chronic inflammatory diseases. The latter, for example, are known to be associated with early and aggressive coronary heart disease, with an earlier average age of onset. Could the age correction justify the different results obtained between incidence and prevalence?
  • the text lacks a comparison table between the IBD population and the general population
  • differences obtained between incidence and prevalence should be commented more comprehensively

Author Response

Response to Reviewers

Dear Editors,

Hereby, we submit our revised manuscript

JCM 1506299: Increased prevalence of myocardial infarction and stable stroke proportions in patients with inflammatory bowel diseases in Quebec in 1996-2015

Petra Anna Golovics, MD, PhD1,2*, Christine Verdon MBBS MSc(A)1*, Panu Wetwittayakhlang, MD1,3,  Christopher Filliter MSc4, Lorant Gonczi MD, PhD5, Gustavo Drügg Hahn MD1,6, Gary E. Wild MD, PhD1, Waqqas Afif MD, MSc1, Alain Bitton MD1, Talat Bessissow MD, MSc1, Paul Brassard MD, MSc4, 7, 8, Peter L. Lakatos MD, PhD 1,5 *

We thank the reviewers for the comments, Please find below the point by point answer to the reviewers’ questions

Reviewer 1

Point 1.  it is not clear how the risk factors (diabetes mellitus, dyslipidemia and hypertension) were defined: with an established diagnosis? by analysis of current therapy? As commented by the authors, the data on smoking is also missing.

Response 1: Thank you for the comment. We were using the health administrative database of the province of Quebec. The more detailed description is available in our previous publication. [5] (Christine Verdon, Jason Reinglas, Janie Coulombe et al. No Change in Surgical and Hospitalization Trends Despite Higher Exposure to Anti-Tumor Necrosis Factor in Inflammatory Bowel Disease in the Québec Provincial Database From 1996 to 2015. Inflammatory Bowel Diseases. Volume 27, Issue 5, May 2021, Pages 655–661.)

The Regie de l’Assurance Maladie du Quebec, a population-based health insurance database for the Province of Quebec, was used. Patients were selected using ICD-9 and ICD-10 definitions for ulcerative colitis (UC) and CD (indeterminate IBD was excluded) in inpatient and outpatient settings and using drug prescriptions from January 1996 to December 2015. Cohort entry corresponded to the IBD diagnosis. Time-to-event was defined as being from the diagnosis of IBD to the minimum date between patients’ first (or second) event, the administrative end of the study (December 2015), or death, whichever occurred first.

For the risk factors we used the cardiovascular guidelines risk factors, and it was identified using the ICD codes.

Unfortunately smoking ICD code is not administered in this administrative database hence we could not assess the importance of the smoking, as discussed in the discussion limitation section. 

Point 2  The distinction between the two cohorts (newly diagnosed IBD cases vs IBD cases with RAMQ prescription insurance  is not clear. Furthermore, the differences between patient characteristics are not commented (a higher incidence of CV risk factors in the second population is evident)

Response 2: The newly diagnosed patients are all the patients with IBD diagnosis between 1996-2015. Those whom have RAMQ prescription insurance are called the insured population. We don’t have access to the private medical insurance data, this reflects missing data on drug prescriptions, but not diagnosis. I added this comment to the 2.2.2. Risk factor analysis section too.

Numerically in the insured population the numbers are higher, but in the 3.2 Risk factors section we explained that we made the calculation separately to this population too and the results were the same as in the total population hence we are not presenting it separately.

Point 3: IBD patients were compared  with a age-matched population. However, the age of onset of the CV disease could be different in the two populations as well as the age of onset of the various risk factors, as is well known in the context of chronic inflammatory diseases. The latter, for example, are known to be associated with early and aggressive coronary heart disease, with an earlier average age of onset. Could the age correction justify the different results obtained between incidence and prevalence?

Response 3: We were using the health administrative database of the province of Quebec for IBD and the available Canadian control population data (rates were available from 2012-2013). To avoid this bias we performed age-stratified subgroup analysis using 2 age categories (<40 and ≥40 year-old patients) and also a gender-stratified analysis (2.2.1 section and 3.2 section).

We identified older age and female gender (p<0.001 for each) as risk factors for developing MI and stroke in both CD and UC in the logistic regression-based prevalence models.

However, the age and gender-standardized prevalence rates for IHD (ischemic heart disease), but not MI were available on the respective website (https://www.canada.ca/en/public-health/services/publications/diseases-conditions/heart-disease-canada-fact-sheet.html)

and the prevalence of IHD in the 30-39 year-olds was 0.4-0.8% in the 40-49 year-olds 1.5-2.9% and 50-59 year-olds 5.1-8.4%, with almost twice as high rates in males compared to females. Of note, IHD is a more broad category, and less reliable overall clinical diagnosis, this  is why we selected the MI to be assessed for cardiovascular risk in IBD and since the IBD group falls mainly in these 3 age group categories the difference in the prevalence rate in IBD between the general population in MI seems real.

Point 4:  the text lacks a comparison table between the IBD population and the general population

Response 4: Thank you for the comment. Of note, we were using the health administrative database of the province of Quebec for IBD and the available Canadian control population data (rates were available from 2012-2013). Thus direct comparative tables are not possible. To increase the strength of the statistical analysis and decrease the risk of bias we performed age-stratified subgroup analysis using 2 age categories (<40 and ≥40 year-old patients) and also a gender-stratified analysis (2.2.1 section and 3.2 section).

Point 5: Differences obtained between incidence and prevalence should be commented more comprehensively

Response 5: Thank you for this comment. This is indeed an important question, and since the incidence rate of MI in IBD was not significantly increased we cannot state that the cardiovascular risk is undoubtedly increased in IBD. This is why we aimed to evaluate the importance of known CV risk factors in the IBD population to assess, which factors can contribute to the CV risk in the IBD patients.

Reviewer 2 Report

Golovics and coworkers in this paper evaluate the cardiovascular and cerebrovascular outcomes and its risk factors in a cohort of patients affected by Inflammatory Bowel Disease (IBD) using a health administrative database of a province of Canada. The manuscript reports the data collected in a very large population (about 36000 IBD patients) of which information regarding cardiovascular outcomes and risk factors are available and suggests a role of IBD as a risk factor of cardiovascular disease. However, several critical methodologic issues, impair the validity of the reported results and seem to not sufficiently support authors’ conclusions.

In detail:

Major issues:

  1. Data on prevalence refers to a period from 1996 to 2015 while the end of follow-up for deriving incidence is not specified. This means that data on prevalence include patients identified about 25 years ago. In this time span, the diagnosis of myocardial infarction has been revised and changed several times. There is any variation in the prevalence of myocardial infarction during the observational period that can the related to diagnostic criteria?
  2. The authors state that they used a reference population for comparisons of prevalence and incidence of cardiovascular outcomes. However, no data is reported regarding this reference population. In addition, the authors should provide, besides the odds ratios, the crude prevalence and incidence data in both populations, and absolute risk estimations should be assessed and reported.
  3. The so-called “comorbidity analysis” appears somewhat misleading regarding the real topic explored. In fact, the role of “classical risk factors on cardiovascular and cerebrovascular disease development is well known, while authors should explore the putative role of IBD. Authors should thus produce a logistic regression (or Poisson regression, when appropriate) using the outcome as the dependent variable and including IBD among predictors. Anyway, the approach that the authors describe remains obscure when they call “univariate” regressions that, actually, are stepwise multiple regression analyses.
  4. Characteristics of the “insured” cohort are substantially different in comparison with the whole population. It should be noted that the insured population cohort suffers from an obvious selection bias that should be addressed. Moreover, a possible interaction bias due to the reported protective effect of anti-TNF biological drugs on cardiovascular outcomes (1–3) should be taken into account.
  5. The authors report an increase in the prevalence but not in the incidence of cardiovascular events in the IBD population. This scenario can occur when a condition is somewhat “protective” allowing a longer survival of patients affected. This is surprising and in contrast with the conclusions drawn by authors regarding the adverse effect of IBD.
  6. Authors report but do not sufficiently comment on the predictive role of the female gender in increasing the prevalence and incidence of cardiovascular events. While the female gender is universally recognized as a protective factor against cardiovascular disease especially in young populations (please provide age distribution of the study population) this result can be simply the expression of the high prevalence of the female gender in the IBD population. The true role of gender should be derived in a regression including IBD among predictors (see point 3)

Minor issues:

  1. STROBE checklist declared in the manuscript but not available in among the supplementary data
  2. Several mistyping errors impair the readability of the manuscript. Please note that the first few statements of the results section are duplicated
  3. Table 1 should include the mean/SD of age and median (IQR) of follow up duration

References

  1. Lee JL, Sinnathurai P, Buchbinder R, Hill C, Lassere M, March L. Biologics and cardiovascular events in inflammatory arthritis: a prospective national cohort study. Arthritis Res Ther 2018;20:171.
  2. Roubille C, Richer V, Starnino T, et al. The effects of tumour necrosis factor inhibitors, methotrexate, non-steroidal anti-inflammatory drugs and corticosteroids on cardiovascular events in rheumatoid arthritis, psoriasis and psoriatic arthritis: a systematic review and meta-analysis. Ann Rheum Dis 2015;74:480–489.
  3. Yang Z-S, Lin N-N, Li L, Li Y. The Effect of TNF Inhibitors on Cardiovascular Events in Psoriasis and Psoriatic Arthritis: an Updated Meta-Analysis. Clin Rev Allergy Immunol 2016;51:240–247.

Author Response

Response to Reviewers

Dear Editors,

Hereby, we submit our revised manuscript

JCM 1506299: Increased prevalence of myocardial infarction and stable stroke proportions in patients with inflammatory bowel diseases in Quebec in 1996-2015

Petra Anna Golovics, MD, PhD1,2*, Christine Verdon MBBS MSc(A)1*, Panu Wetwittayakhlang, MD1,3,  Christopher Filliter MSc4, Lorant Gonczi MD, PhD5, Gustavo Drügg Hahn MD1,6, Gary E. Wild MD, PhD1, Waqqas Afif MD, MSc1, Alain Bitton MD1, Talat Bessissow MD, MSc1, Paul Brassard MD, MSc4, 7, 8, Peter L. Lakatos MD, PhD 1,5 *

We thank the reviewers for the comments, Please find below the point by point answer to the reviewers’ questions

Reviewer 2

1.Major issues:

Point 1: Data on prevalence refers to a period from 1996 to 2015 while the end of follow-up for deriving incidence is not specified. This means that data on prevalence include patients identified about 25 years ago. In this time span, the diagnosis of myocardial infarction has been revised and changed several times. There is any variation in the prevalence of myocardial infarction during the observational period that can the related to diagnostic criteria?

Response 1: Thank you for the question. The end of follow-up for deriving the incidence rates was the same (2015). We agree with the comment that the diagnosis of the MI changed over the observation period, however this does not affect the method of identifying IBD patients with MI in the RAMQ database. This is an administrative data base thus the use of the respective ICD codes were captured to identify the MI event as given by the primary treating physician and this was not modified by our data capture.

We used the following ICD codes for MI (ICD-10) - I21-I22. (ICD-9) – 410 and for Ischemic heart disease (includes MI) (ICD-10)- I20-25; (ICD-9): 410-414. In case of stroke (ICD-10):  I63-64. (ICD-9) 433-438, if stroke includes subarachnoid and intracerebral haemorrhage then: (ICD-10) I60-I64 (ICD-9) 431-438.

Point 2:  The authors state that they used a reference population for comparisons of prevalence and incidence of cardiovascular outcomes. However, no data is reported regarding this reference population. In addition, the authors should provide, besides the odds ratios, the crude prevalence and incidence data in both populations, and absolute risk estimations should be assessed and reported.

Response 2: Thank you for the comment, the administrative database derived IBD cohort was compared to the available Canadian control population data (rates were available from 2012-2013), we added the link to the Government of Canadian website responsible for the collection and publication of rates in the general population (see reference and link in the Methods)

Point 3: The so-called “comorbidity analysis” appears somewhat misleading regarding the real topic explored. In fact, the role of “classical risk factors on cardiovascular and cerebrovascular disease development is well known, while authors should explore the putative role of IBD. Authors should thus produce a logistic regression (or Poisson regression, when appropriate) using the outcome as the dependent variable and including IBD among predictors. Anyway, the approach that the authors describe remains obscure when they call “univariate” regressions that, actually, are stepwise multiple regression analyses.

Response 3: Thank you for the comment, indeed we performed a different analysis than what the reviewer was asking for. Since the prevalence, but not the incidence of the MI risk was increased in IBD, we felt that the CV risk- at least according to our data- was not unquestionably higher, while the risk of stroke was not different from the background population. Therefore our aim was to assess the importance of conventional risk factors in IBD and to evaluate how much they contribute to the CV risk in IBD. Moreover, we did not have the distribution of all risk factors in the general Canadian population. See also Reviewer 1, point 5

Point 4:  Characteristics of the “insured” cohort are substantially different in comparison with the whole population. It should be noted that the insured population cohort suffers from an obvious selection bias that should be addressed. Moreover, a possible interaction bias due to the reported protective effect of anti-TNF biological drugs on cardiovascular outcomes (1–3) should be taken into account.

Response 4: We agree with the comment of the reviewer, thus is why we already addressed this bias in the discussion in the limitation part “In addition, the Quebec drug plan for biologics therapy is two-fold, with public RAMQ or private insurance coverage; the latter was not captured in our data set. Of note, 41.8% of prescribed drug spending is paid for by provincial healthcare plans across Canada.[17]”

Moreover, in the results we mention that we performed the risk factor analysis for the total and the insured population separately (as a sensitivity analysis) and that there was no difference.

In addition, we discussed in the 3.3. section the importance of the disease severity. Although there is no direct data on disease severity, one may expect that patients on biologic therapy identifies patients with a higher inflammatory burden and this more severe IBD phenotype and systemic inflammatory burden is responsible for the higher MI incidence. We also added your reference suggestion to the introduction however all those studies analyzed in rheumatoid arthritis, psoriasis, psoriatic arthritis and ankylosing spondylitis. No earlier data is available for IBD.

We also agree that the use of the anti-TNF therapies exerts an anti-inflammatory effect and hence may be directly protective against CV outcome. However, to prove this requires an enormous effort, optimally two large severe IBD patient cohorts with similar age, gender (preferentially, 50+ age-group to allow higher event count and similar conventional risk factor patterns) distribution one being exposed to anti TNF therapy, while the other not, should be followed-up for several year and this is beyond the scope of the present study

Point 5: The authors report an increase in the prevalence but not in the incidence of cardiovascular events in the IBD population. This scenario can occur when a condition is somewhat “protective” allowing a longer survival of patients affected. This is surprising and in contrast with the conclusions drawn by authors regarding the adverse effect of IBD.

Response 5: Thank you for this comment. This is indeed an important question, and since the incidence rate of MI in IBD was not significantly increased we cannot state that the cardiovascular risk is undoubtedly increased in IBD, while the risk of stroke was not increased. Partly in contrast to previous studies and explaining why we finish with a conservative conclusion.

In addition, this is why we evaluated the importance of known CV risk factors in the present IBD population to assess, which factors can contribute to the CV risk in the IBD patients.

Point 6 : Authors report but do not sufficiently comment on the predictive role of the female gender in increasing the prevalence and incidence of cardiovascular events. While the female gender is universally recognized as a protective factor against cardiovascular disease especially in young populations (please provide age distribution of the study population) this result can be simply the expression of the high prevalence of the female gender in the IBD population. The true role of gender should be derived in a regression including IBD among predictors (see point 3)

Response 6: Thank you for the comment. The finding of the female gender as CV risk factor is not unique to our study.Kirchgesner et al reported also that women with IBD compared to men with IBD were at higher risk for in all arterial disease groups in both CD and UC with the highest risk in women with CD.

In the present study, the prevalence of females was 53.9%, which is not increased to the Canadian general population.

In addition, the calculated prevalence rates in IBD in males were 5.1% vs in the Canadian background population 2.9% (OR= 1.8, p<0.0001), while in IBD in females the rate was 3.00 vs the Canadian background rate was 1.1% (OR= 2.8, p<0.0001). Similarly in the incidence analysis, using a Poisson model, the MI incidence in IBD in males was 0.301 vs in the Canadian population 0.310 (p=0.593), while in females the MI incidence in IBD was 0.180 vs in the Canadian population 0.15 (p=0.007 , IRR=1.2, 1.05-1.37)

This suggests that the protective effect of female gender to CV outcomes is diminished in IBD and is probably counteracted by the presence of chronic inflammatory burden of the disease, and this effect may be more pronounced in females compared to males (we added a short comment also to the discussion)

Minor issues:

Point 7:  STROBE checklist declared in the manuscript but not available in among the supplementary data

Response 7 Thank you, we corrected and uploaded now.

Point 8: Several mistyping errors impair the readability of the manuscript. Please note that the first few statements of the results section are duplicated

Response 8: Thank you, we corrected.

Point 9 Table 1 should include the mean/SD of age and median (IQR) of follow up duration

Response 9: This is how data was provided by the administrative data base. Unfortunately we can not present in another format. Similar data was presented in an earlier paper from our group on the same cohort (Christine Verdon, Jason Reinglas, Janie Coulombe et al. No Change in Surgical and Hospitalization Trends Despite Higher Exposure to Anti-Tumor Necrosis Factor in Inflammatory Bowel Disease in the Québec Provincial Database From 1996 to 2015. Inflammatory Bowel Diseases. Volume 27, Issue 5, May 2021, Pages 655–661.)

References

  1. Lee JL, Sinnathurai P, Buchbinder R, Hill C, Lassere M, March L. Biologics and cardiovascular events in inflammatory arthritis: a prospective national cohort study. Arthritis Res Ther 2018;20:171.
  2. Roubille C, Richer V, Starnino T, et al. The effects of tumour necrosis factor inhibitors, methotrexate, non-steroidal anti-inflammatory drugs and corticosteroids on cardiovascular events in rheumatoid arthritis, psoriasis and psoriatic arthritis: a systematic review and meta-analysis. Ann Rheum Dis 2015;74:480–489.
  3. Yang Z-S, Lin N-N, Li L, Li Y. The Effect of TNF Inhibitors on Cardiovascular Events in Psoriasis and Psoriatic Arthritis: an Updated Meta-Analysis. Clin Rev Allergy Immunol 2016;51:240–247.

Round 2

Reviewer 2 Report

  1. (Point 1). I agree that ICD codes did not change during the observation period. What changed is the sensitivity in the identification of myocardial necrosis since the introduction in the clinical practice of high-sensitivity troponins. It is a common perception that this clinical tool actually reduced the incidence of diagnoses of (unstable) angina while increasing the diagnoses of NSTEMI. Surprisingly, this phenomenon was not captured by the administrative registry that you analyzed. Please add a comment in the discussion section
  2. (Point 2). Thank you for your answer. In the text (page 5) please indicate the reference unit for incidence (e.g. x 1000 x year)
  3. (Point 3). Your statistical approach to the research question is still obscure to me. In the presented analyses you can only catch the relative association (not causative role) of each of the risk factors in the IBD population with the endpoints analyzed. Without any formal comparison with the not-IBD population, you cannot determine if there is any different contribution of these factors in comparison with not-IBD subjects. This is more striking when you should comment on the different behavior of the female gender as a risk factor in this population in comparison with the general population.
  4. (Point 4). Thank you for your response
  5. (Point 5). I appreciate that the authors agree with my comment. However, following the analyses performed you can only conclude that IBD patients share the same cardiovascular risk factors with the general population with the only exception of the female gender, which here appears as an additional risk factor. This should produce an increased global risk for myocardial infarction, but this is not the case (see data on incidence). Please note that if we consider the opposite role of the female gender in the IBD population in comparison to the general population, the same incidence of myocardial infarction can be explained only with a lesser effect of one or more of the other classical risk factors. This point can be resolved only with an appropriate statistical approach (see also point 3).
  6. (Point 6). As for points 3 and 5.
  7. (Point 7). Thank you for updating the required information
  8. (Point 8). Thank you for your corrections
  9. (Point 9). Thank you for your response. This is a major limitation of this study.

Author Response

Response to Reviewers

Dear Editors,

Hereby, we submit our revised manuscript

JCM 1506299

Increased prevalence of myocardial infarction and stable stroke proportions in patients with inflammatory bowel diseases in Quebec in 1996-2015

Petra Anna Golovics, MD, PhD1,2*, Christine Verdon MBBS MSc(A)1*, Panu Wetwittayakhlang, MD1,3,  Christopher Filliter MSc4, Lorant Gonczi MD, PhD5, Gustavo Drügg Hahn MD1,6, Gary E. Wild MD, PhD1, Waqqas Afif MD, MSc1, Alain Bitton MD1, Talat Bessissow MD, MSc1, Paul Brassard MD, MSc4, 7, 8, Peter L. Lakatos MD, PhD 1,5 *

We thank the reviewers for the comments, Please find below the point by point answer to the reviewers’ questions

Major issues:

(Point 1). I agree that ICD codes did not change during the observation period. What changed is the sensitivity in the identification of myocardial necrosis since the introduction in the clinical practice of high-sensitivity troponins. It is a common perception that this clinical tool actually reduced the incidence of diagnoses of (unstable) angina while increasing the diagnoses of NSTEMI. Surprisingly, this phenomenon was not captured by the administrative registry that you analyzed. Please add a comment in the discussion section

Thank your comment, we included this now in the discussion

It is a common perception there was change the sensitivity in the identification of myocardial necrosis since the introduction in the clinical practice of high-sensitivity troponins and this may reduce the incidence of diagnoses of (unstable) angina while increasing the diagnoses of NSTEMI. Of note, this phenomenon was not captured in the present administrative registry”

(Point 2). Thank you for your answer. In the text (page 5) please indicate the reference unit for incidence (e.g. x 1000 x year)

Thank your comment, we modified data presentation as requested

(Point 3). Your statistical approach to the research question is still obscure to me. In the presented analyses you can only catch the relative association (not causative role) of each of the risk factors in the IBD population with the endpoints analyzed. Without any formal comparison with the not-IBD population, you cannot determine if there is any different contribution of these factors in comparison with not-IBD subjects. This is more striking when you should comment on the different behavior of the female gender as a risk factor in this population in comparison with the general population.

Thank your comment, I believe that there is no disagreement between what we say and the reviewer’s interpretation. In contrast, we fully agree with the reviewer’s interpretation, in fact our goal was exactly to study the “relative“ importance of the “conventional” CV risk factors in the IBD patients. In other words, we aimed to study only if the conventional CV risk factors are relatively important (or not) in IBD, unrelated to the increased or not prevalence/incidence data. Thus we fully agree, that this analysis shows only that IBD patients with the conventional CV risk factors are at risk for the outcome(s) and NOT whether these are more or less common in IBD vs non-IBD controls,but this was again the aim of the present study.

We also agree that the study of IBD and non-IBD controls can evaluate whether the prevalence of common CV risk factors are increased in IBD vs controls, yet this was not the aim of the study to assess the prevalence of risk factors compared to the general population, moreover the incidence was also not increased. In addition, in the literature, the published data are conflictive on the presence of CV risk in IBD (eg Dorn et al Am J GE 2007, metaanalysis) and our paper is more in line with these, questioning a clinically important association. Therefore, we believe that the above analysis in the present cohort would be negative, and would prove a lack of generally increased prevalence of conventional CV risk factors in IBD

In addition, the investigation of complex CV risk factors in IBD and non-IBD populations would be a complete and independent project on its own. The very few publications that tried to estimate this appropriately, reported that the complex risk assessment could not be assessed reliably (Cardiovascular risk assessment and treatment in chronic inflammatory disorders in primary care. Emanuel G, Charlton J, Ashworth M, Gulliford MC, Dregan A. Heart. 2016 Dec 15;102(24):1957-1962. doi: 10.1136/heartjnl-2016-310111. Epub 2016 Aug 17), but there seemed to be an increased frequency of the CV risk factors, although significantly decreasing over time

This clarification is now included in the revised discussion to better clarify the aim and interpretation of the risk factor analysis and also to explain the uncertainty of the assessment of common CV risk factors in IBD

(Point 5). I appreciate that the authors agree with my comment. However, following the analyses performed you can only conclude that IBD patients share the same cardiovascular risk factors with the general population with the only exception of the female gender, which here appears as an additional risk factor. This should produce an increased global risk for myocardial infarction, but this is not the case (see data on incidence). Please note that if we consider the opposite role of the female gender in the IBD population in comparison to the general population, the same incidence of myocardial infarction can be explained only with a lesser effect of one or more of the other classical risk factors. This point can be resolved only with an appropriate statistical approach (see also point 3).

See earlier point for the detailed answer, and yes, we fully agree with the reviewer’s conclusion, that IBD patients WITH the event are the ones with the increased prevalence of CV risk factors, yet again our aim was exactly to perform the relative analysis in the IBD population, which identifies those at risk among the IBD patients, but this does not necessarly equals an increased overall CV risk or an increased incidence and/ or prevalence

The paper was once more proofread by native English speaker and linguistic errors and typos were corrected.

We thank again the Reviewers for their important questions and comments, and we hope that the revised version is improved and the answers are acceptable especially regarding the relative CV risk assessment and that the paper can be of value to the readers of the Journal of Clinical Medicine

Petra Golovics, Panu Wetwittayakhlang and Peter L Lakatos

on behalf of the authors

Peter L LAKATOS, MD, PhD, DSc, FEBG, AGAF
Director of IBD Centre, Professor of Medicine
McGill University Health Centre
, Montreal General Hospital,
1650 Ave. Cedar, D7-201,Montreal, QC, H3G 1A4

Tel:  +-1-514-9341934 x ext 45567

Fax: +1-514-934-4452 
Tel: +1-514-4317994

email: [email protected]

Round 3

Reviewer 2 Report

The authors deeply revised the manuscript and appropriately addressed all the raised issues. I acknowledge their efforts and appreciate how they amended the manuscript. I just point your attention to two unremarkable typing errors:

  1. Row 211: the incidence is incorrectly reported (0.122/1000006yr instead of 0.122/100000/yr)
  2. Row 329: the reference is missing (Error! Bookmark not defined.)